# Augmenting Language Models With Composable Differentiable Libraries

## Abstract

Important reasoning tasks such as planning are fundamentally algorithmic, meaning that solving these tasks robustly requires inducing the underlying algorithms, rather than shortcuts. Large Language Models lack true algorithmic ability primarily because of the limitations of neural network optimization algorithms, their optimization data, and optimization objective, but also due to the inexpressivity of the transformer architecture. To address this lack of algorithmic ability, our paper proposes augmenting LLMs with an internal reasoning module. This module contains a library of fundamental operations and sophisticated differentiable programs so that common algorithms do not need to be learned from scratch. To accomplish this, we add memory, registers, basic operations, and adaptive recurrence to a billion-parameter scale transformer architecture built on LLaMA3.2. Then, we define a method for directly compiling algorithms into a differentiable starting library, which is used natively and propagates gradients for optimization. In this paper, we study the feasibility of this augmentation by fine-tuning an augmented LLaMA 3.2 on simple algorithmic tasks with variable computational depth, such as a recursive Fibonacci algorithm or insertion sort.

## 1 Introduction

Machine learning is relaxed program induction, where, implicitly or explicitly, the goal is to find programs that accomplish a given task [1, 2, 3, 4]. For example, a large language model trained on math problems must implicitly learn a calculator program internally. Furthermore, models may aim to induce more complex internal programs, such as sorting algorithms, planning algorithms, or combinatorial solvers. However, gradient descent has no guarantee of recovering such programs, and often approximates them via statistical features and other shortcuts [5, 6]. To avoid the issue of inducing already-known programs from data, we use *neural compilation*, which compiles code into neural network parameters [7, 8, 9, 10]. Specifically, we augment a large language model with a compiled library of differentiable programs, which can be used as a foundation for further learning. Ideally, the model will learn *compositions* of subprograms in the library [11].

Language models are optimized to model natural language through objectives like masked-token or next-token prediction. In theory and practice, these objectives are insufficient for the emergence of authentic reasoning ability, even when it may appear superficially [12, 5, 6, 13, 14]. In general, this lack of reasoning ability is a fundamental flaw that is not easily mitigated via prompting or fine-tuning [15, 12]. First, algorithmic reasoning, by definition, requires an architecture to be universally expressive. Second, optimization must be able to find target programs. However, transformer expressivity is upper-bounded by $TC^0$ [16], meaning that a single transformer pass can only at best approximate algorithms using a highly-parallel circuit [5]. Furthermore, there is ample empirical evidence that optimization does not recover even programs within $TC^0$ [12, 6].

Augmentation aims to address the limitations of large language models. For instance, a language modeling objective is often insufficient to induce a robust calculator sub-program, so it is common to augment a language model with a calculator. Even when appropriate tools are available, a model must use them correctly, by providing the right inputs to the right tool in the right context, which we call the parsing/selection problem. Tool use is often approached via prompting, fine-tuning, or bootstrapping [17]. Differentiability is advantageous for fine-tuning, as it allows supervising on answers rather than tool inputs, and allows for tighter integration than external tool augmentation.

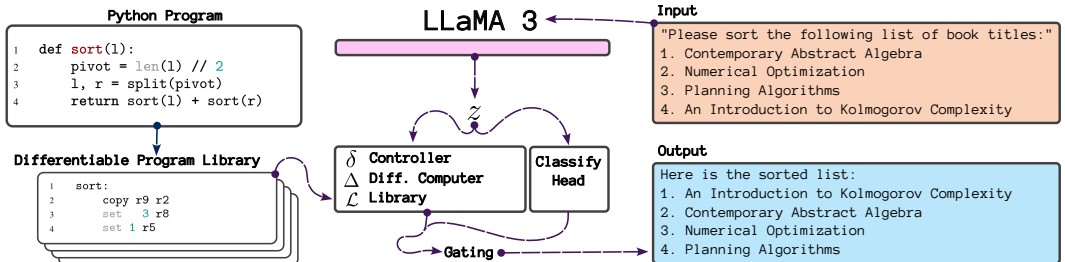

Figure 1: The proposed architecture, which adds a library of differentiable programs to LLaMA3. In principle, these programs can be composed to flexibly complete new tasks without re-learning individual subprograms. For example, the model may first parse text, sort a list, and then count the characters in the first title by composing `sort()` and `count()` with the LLM's parsing ability.

We augment LLaMA 3.2 with a differentiable computer that runs after the primary model, as depicted in Figure 1. This computer contains a library of multiple compiled programs, as well as primitive operations such as arithmetic. In principle, this augmentation makes the overall model universal *in a single pass*. Because the computer is differentiable, it's potentially possible to adapt the overall model to use the compiled library functions for new tasks, or to parse their inputs from natural language. Accordingly, we perform experiments that test the impact of computation depth on training, as well as experiments which require the model to select and compose library functions in-context. Overall, we find that this augmentation works best for highly parallel and shallow algorithms, but is promising in terms of enabling foundation models to do algorithmic reasoning.

## 1.1 NEURAL COMPILATION

Neural compilation is a technique for deterministically transforming code into neural network parameters that express the exact same program in a given architecture. Precursors to neural compilation were first discussed in Siegelmann and Sontag, and then implemented in Gruau et al. [18, 7]. However, the first adaptive (trainable) neural compilation technique was first defined in Bunel et al. [8]. Similarly, there are modern approaches to neural compilation, based on the transformer architecture, but these either focus on interpretability, are not universal, or are not adaptive [19, 9, 20, 16, 10].

## 2 RELATED WORK

### 2.1 PREVIOUS NEURAL COMPILATION TECHNIQUES

**Adaptive Neural Compilation** augments a recurrent neural network with memory, registers, and a differentiable interpreter for a minimal assembly language [8]. Then, [8] compiles algorithms by solving for weights analytically. This model relied on a lookup-table based ALU, unit vector numeric encodings, dot-product memory/register lookups, and probability mixtures for control flow. This work focused on learning contextual programs (e.g. sorting biased lists), but in contrast we focus on compilation as a means to specify algorithms to Large Language Models.

**RASP/Tracr/CoNN** describe a neural compilation technique for unaugmented transformers, aimed at interpretability. Specifically, RASP defines a minimal language [19], Tracr defines a working compiler [9], and CoNN exploits the Tracr compiler to augment a transformer. While CoNN compiled addition and subtraction, their mixture-of-experts approach has a basic calculator directly output the answer as a series of tokens, which is limited only to very simple problems and does not support compositionality or training for new tasks [21]. In comparison, our work is the first to experiment with end-to-end trained large language models augmented with universal programs.

**Looped Transformer** constructs a universal machine within a recurrent transformer architecture. However, it is not intended to be adaptive, nor is it explicitly constructed for library learning or integration with pretrained LLMs [10].

## 2.2 Differentiable Computing and Program Synthesis

**Differentiable computing** is the idea that programs can be approximated by neural networks by defining differentiable primitives that support universal computation, for instance by using softmax attention to simulate array access. Recurrent neural networks and LSTMs are early instances of differentiable computers, and generally performed well for several decades, but in the limit cannot learn and generalize arbitrary programs from data [22]. One potential reason for these failures is a lack of inductive bias via expressive primitives, but the critical reason is optimization difficulty [23].

**Neural Turing Machines** [24, 25] construct a sophisticated differentiable computer, and demonstrate its application to complex tasks, such as inducing sorting, planning, or graph algorithms. NTMs are foundational to differentiable computing, however, they are exceptionally hard to train, even in comparison to RNNs and LSTMs [26]. This raises possibility of architectures which achieve both the expressiveness of NTMS and the trainability, parallelism, and capacity of transformers [27].

**Graph Neural Networks** are a successor to Neural Turing Machines specialized in expressing graph algorithms [28, 29, 30, 31, 32, 33]. In practice, graph neural networks can be more trainable than NTMs. However, like any method which relies on gradient descent for induction, there are no hard guarantees and generalization is not perfect, even if overall performance is improved [34]

**Program Synthesis** is closely related to differentiable computing, and studies the practice of inducing code from specifications [35, 3, 4, 1]. Generally this entails symbolic search, with an emphasis on addressing combinatorial explosion via heuristics or pruning. However, program synthesis also overlaps with differentiable computing significantly, and neural networks are often used as generators or heuristics of programs [36, 11, 37, 38, 39, 40, 41].

**Sketching** improves program synthesis by providing a human-specified template of the desired output, and having an algorithm fill in this template [35, 42, 43]. Contextual programs in [8] and our initial library share heritage and draw inspiration from sketching.

**Library learning** focuses on organizing acquired skills for compositional reuse [11, 44]. By creating abstractions, learning more complex algorithms ideally becomes a matter of recomposing library skills, rather than learning from scratch. Beyond abstractions found by a learning algorithm, this paper aims to provide a *foundation* of abstractions, which are compiled into the starting library. This foundation can range from simple arithmetic operations to fully defined planning algorithms. In either case, the goal is to provide an inductive bias for reliably learning algorithmic tasks.

## 2.3 Large Language Model Tool Use

**LLM Tool Augmentation** has been explored as an alternative method to improve reasoning ability in neural networks. For instance, models like GPT have been augmented with calculators or Python interpreters [45] via a text interface. The primary difference between neural compilation and typical LLM tool use is that because they are differentiable, neurally compiled components require fewer intermediate labels and support fine-tuning with an integrated tool. Still, differentiability alone is not a guarantee that correct tool use behavior will be learned from answer supervision, so we do not propose replacing conventional approaches to tool use entirely.

**Augmenting LLMs with Neural Algorithmic Reasoners** Graph neural networks (GNNs) are promising for completing algorithmic reasoning tasks, such as those defined in the CLRS Algorithmic Reasoning Benchmark [34]. Similar to the proposition of this paper, graph neural networks are promising as a potential augmentation to Large Langauge Models. In particular, [46] explores using a neural algorithmic reasoner (NAR) to augment the Chinchilla large language model, creating what they call a TransNAR.

This approach relies on generating synthetic data using a known algorithm, and training an NAR to approximate the source algorithm. Then, the overall augmented model (the TransNAR) receives both text and a structured graph input, and produces a text output. The NAR correctly handles the algorithmic aspect of the task, and shares an embedding with the overall transformer, enabling the overall model to reliably answer reasoning questions.

While effective in many ways, this approach has two downsides compared to our proposal: The source algorithms must be approximated via optimization, which can be reliable (e.g. learning an

algorithm with 99% accuracy *in distribution*), but doesn't carry guarantees like direct neural compilation does. Also, this optimization process is computationally expensive compared to analytical compilation, and *both* require having access to the source algorithm. Furthermore, generating the synthetic training data requires making a specialized version of the source algorithm that provides intermediate hints. Also, the TransNAR is provided with a pre-parsed graph, which skips the important problem of parsing natural language into an appropriate structure. As we will see later in the paper, a significant source of difficulty in using LLMs for reasoning tasks is that their intermediate representations are not structured.

## 2.4 GRADIENT ESTIMATION

An alternative to making specialized differentiable machines is to instead treat tools as "black boxes", and use gradient estimation techniques to learn how to use them. For example [47, 48] allows a black-box combinatorial solver to be used as a differentiable layer in a neural network. These approaches are valuable, but can potentially be sample-inefficient or produce noisy gradients.

## 3 AUGMENTING LLaMA 3.2 WITH A DIFFERENTIABLE LIBRARY

Our model augments the LLaMA 3.2 transformer architecture with a differentiable computer, $\Delta$, and associated program library $\Lambda$. Fundamentally, an intermediate layer of the transformer provides inputs to the differentiable computer (as one-hot classifications) and selects programs to run. The differentiable computer $\Delta$ is based on the register machine introduced in Bunel et al. [8], except that it has been expanded to allow for the program library $\Lambda$ to be accessed via a special `call` instruction. $\Delta$ interprets a set of assembly instructions $A$. A program $\rho$ is a sequence of these instructions, and the library $\Lambda$ is a collection of programs. The computer has state $S$ in the form of memory $M$ and registers $R$, and tracks execution with an instruction counter $c$ and halting probability $h$.

$$S = (M, R, c, h) \tag{1}$$

### 3.1 LIBRARY STRUCTURE

The fundamental contribution of this paper is augmenting an LLM with a differentiable standard library of programs. The overall model uses the program library by selecting programs and inputs to run. Accordingly, composing library functions for new tasks becomes a matter of selecting the appropriate combination of functions to run and providing their parsed inputs.

**Parsing/Selection Problem**    The overall model must *parse* a given natural language input, and provide appropriate inputs for $\Delta$ in the form of an initial state. Then, the model must *select* a program $\varrho$. In our experiments (Section 4), we first study parsing/selection in an isolated form, where a minimal model learns the correct permutation for a task, and then we study the full parsing/selection problem in the context of transformer models with natural language inputs.

**Call Instruction**    Creating a differentiable library fundamentally relies on introducing a method for calling functions arbitrarily. To achieve this, we add a `call` primitive, supported by a `store` instruction, which stores the current program counter in a given register. Doing this allows returning from functions by designating a special return address register. The `call` primitive simply runs `store` and moves the instruction counter into the new function, and the called function returns to the stored location when finished.

### 3.2 MODEL BACKGROUND

**Differentiable Memory**    Bunel et al. defines differentiable memory as a matrix $M_{ij}$, where the dimension $i$ is an address and the dimension $j$ is an encoding. An address $a$ is a unit vector, produced via softmax output. Reading from memory at an address is done with the dot product:

$$r_j = M_{ij}a_i \tag{2}$$

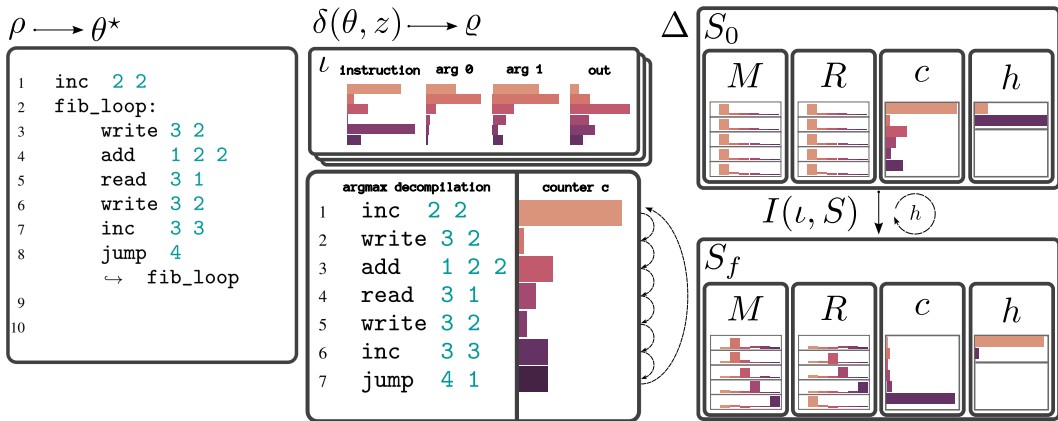

Figure 2: Differentiable Register Machine, Introduced in Bunel 2016 [8]

Writing a vector to memory requires updating all of memory using probability mixtures. First, for an address $a$, vector $c$ being written to $a$, and overall probability of writing, $p$, a memory update is:

$$M_{\text{write}} = (1 - a) \odot M_{\text{old}} + a \otimes c \tag{3}$$

$$M_{\text{new}} = (1 - p) \odot M_{\text{old}} + p \odot M_{\text{write}} \tag{4}$$

$(1 - a) \odot M$ represents kept (unaltered) memory content, and $a \otimes c$ represents new, written content.

**Differentiable Registers**  Registers are defined as a matrix $R_{ij}$. To write an output $c$ to address $a$:

$$R_{ij} = (1 - a) \odot R_{ij} + a \otimes c \tag{5}$$

Reading from an address $a$ to a value $v$ is done with a dot product: $v_j = R_{ij} a_i$. The distinction between memory and registers is that instruction inputs/outputs use registers, not memory. Also, registers are always written at every timestep by any instruction, while memory is only written from `read` or `write` instructions when they have non-zero probability.

**Differentiable Register Machine**  The computer executes a set of assembly instructions, $A$, representing the computer's language. A differentiable program $\varrho$ is structured as a list of these instructions and their arguments, where each instruction can be accessed at its address. These addresses are tracked via a special instruction counter, $c$. Then, a differentiable interpreter $I$ runs instructions $A$ in order to execute the program.

There are two special instructions necessary: `jump` and `halt`, which control program flow. The `jump` instruction takes two inputs: a register holding a conditional flag, and a register holding a program address. If the conditional flag is true (equal to 1), then the program jumps to the new program address. Finally, the `halt` instruction simply finishes the control flow, without executing the remainder of the program. Since this instruction is probabilistic, it is thresholded when executing programs in practice. A particular probabilistic instruction $\iota$ is a multinomial distribution over all possible instructions in $A$. Accordingly, each instruction has an individual probability, and in particular we denote the special scalar portions of $\iota$ as $h$, $j$, and $w$ for the components representing halting, jumping, and writing probabilities.

**Probabilistic Execution**  Instructions, the program counter, and addresses are represented as multinomial probability distributions output by softmax. Accordingly, the program and interpreter is always in *superposition*. Instead of running a single instruction at a time, the interpreter runs *everything, everywhere, all at once*, but with execution and results weighted by the distributions for instructions, program counters, and addresses. In the case that every distribution is dirac-delta (100% probability of one possibility), then execution is fully deterministic. See Figure 1.

Program execution is tracked as a probability mixture between incrementing the instruction counter and jumping to a new location $l$ in the program, based on the condition probability $p$:

$$c_{t+1} = (1 - j) \cdot \underbrace{\mathrm{inc}(c_t)}_{\text{next line}} + j \cdot \underbrace{\left((1 - p) \cdot \mathrm{inc}(c_t) + p \cdot l\right)}_{\text{jump destination}} \tag{6}$$

**Differentiable Interpreter**   An interpreter $I : S_t, \iota \mapsto S_{t+1}$ runs each instruction by querying a 4D lookup table. This model is based on one-hot encodings of size $n$, so this lookup table $T$ has dimensions $|A| \times n \times n \times n$. This table is filled according to each instruction, with special cases for reading or writing to registers/memory. For instance $T_0, :, :, :$ is the addition table. To run a instruction, first the register values are resolved to $u_j, v_j = R_{ij} r_i \forall r$. The final lookup is:

$$o_l = T_{ijkl} f_i u_j v_k \tag{7}$$

Intuitively, this corresponds to first looking up a particular operation (e.g. add), then the first argument (e.g. 2), and then the second argument (2) to get the answer 4. Each instruction specifies a register to store the output in, so finally $o_l$ is written to $R$ using equation 5. Since each operation $f$ and argument $r$ are independent multinomial distributions, this entire operation is probabilistic, so the output $o$ is potentially a mixture of running different instructions with different registers.

## 3.3   MODEL COMPONENTS: ARITHMETIC

**Differentiable Lookup Tables**   Tables trade memory for computation by pre-calculating the answers to input combinations. Intuitively, these take similar form to grade-school arithmetic tables (right). To access a lookup table differentiably, one-hot encoded unit vectors are used as indices for lookup via sequential dot products. For instance the number 2 encodes to $[00100]$ for encoding $n = 5$. A dot product in one axis is equivalent to selecting the row or column containing 2, e.g. $[02468]$. If the other operand is 4 ($[00001]$), then a second dot product selects the final element, 8, which is the answer to $2 \times 4$, the two index vectors. In practice, answers in a lookup table such as this are encoded using unit vectors, making a 3D tensor $M_{ijk}$ where the axes $i$ and $j$ correspond to the first operands, and $k$ is the encoding dimension of the answer. Then, a lookup is an einstein summation identical to equation 7.

| $\times$ | 0 | 1 | 2 | 3 | 4 |
|---|---|---|---|---|---|
| 0 | 0 | 0 | 0 | 0 | 0 |
| 1 | 0 | 1 | 2 | 3 | 4 |
| 2 | 0 | 2 | 4 | 6 | 8 |
| 3 | 0 | 3 | 6 | 9 | 12 |
| 4 | 0 | 4 | 8 | 12 | 16 |

A grade-school multiplication table, encoded differentiably for modulo 5:

$$\begin{bmatrix} 00001 & 00001 & 00001 & 00001 \\ 00001 & 00010 & 00100 & 01000 \\ 00001 & 00100 & 10000 & 00010 \\ 00001 & 01000 & 00010 & 10000 \end{bmatrix}$$

**Differentiable Circuits**   An alternative to lookup tables is to encode basic operations via circuits. This is done by defining differentiable relaxations of common logic gates, and then building conventional circuits, such as the ripple-carry addition circuit, from them. This has been explored several times in previous literature, and there are multiple options for defining differentiable logic gates with different trade-offs [49]. We opt for probabilistic interpretations of and, or, not and xor:

$$\mathrm{and(a, b)} = a \cdot b \qquad\qquad \mathrm{or(a, b)} = a \cdot b + (1 - a) \cdot b + a \cdot (1 - b) \tag{8}$$

$$\mathrm{xor(a, b)} = (1 - a) \cdot b + a \cdot (1 - b) \qquad \mathrm{not(a)} = 1 - a \tag{9}$$

Making larger differentiable circuits is a simply a matter of re-defining classical circuits with these differentiable gates. Accordingly, we define differentiable circuits for addition, multiplication, subtraction, and long division. Compared to lookup tables, circuits require minimal storage space and generalize indefinitely. However, they require binary representations and have long gradient paths.

## 3.4   MODEL AUGMENTATION AND TRAINING

**Augmentation**   Our model augments LLaMA 3.2 in the final layer, as represented in Figure 1. Specifically, we focus on a regime where the LLM receives tokenized natural language, and runs all but the last layer to produce an intermediate vector, $z$. For instance, in the case of LLaMA 3.2 1B, $z$ is a length 2048 vector. $z$ is then used to produce inputs to the differentiable computer, namely classifying which library function to call and what inputs to provide to it. These classifications take

the form of the initial state of the differentiable computer, namely a tuple $(M, R, c, h)$ which is represented as a $n \times n \times 2$ tensor, $c$ is initialized to the encoding of $0$, and $h$ is initialized to $0$.

Furthermore, the augmentation layer *selects* a target algorithm as a categorical classification, meaning that all library functions are run, but are weighted by this softmax output. Because of this, *all of the library functions are used during training, meaning that it is potentially possible to learn selection from answer supervision.* The final layer of the augmented model takes the differentiable computer's output, and produces a final answer, for instance in the form of an integer (for arithmetic and Fibonacci) or a sorted list. This answer is supervised by cross-entropy loss. In future work, we will consider alternate augmentations, such as one that occurs in the middle layers of a model or especially one where an intermediate `main()` function is synthesized, which would better support compositionality and integration.

## 4 EXPERIMENTS

### 4.1 PRELIMINARY STUDIES WITH A MINIMAL NEURAL NETWORK

Before scaling to billion-parameter models, we explore behaviors of components of our differentiable computer, namely lookup tables, circuits, and small programs. These experiments use a minimal neural network with one layer before the computer and one layer after, with inputs as one-hot encodings rather than tokenized text. These networks simply need to route inputs/outputs correctly to/from the computer, which is replaced by parsing in case of LLMs. We find that lookup tables are more learnable than circuits, and that we can learn recursive algorithm routing to a certain depth.

**Bootstrapping From Learnability to Generalization**
In this experiment, we augment a minimal network with either a differentiable lookup table or a differentiable arithmetic circuit. Differentiable circuits using binary representations are ideal for scaling arithmetic to arbitrary numbers. However, lookup tables based on one-hot encodings are far more trainable. Our experiments (Figure 3) confirm these hypotheses: Lookup tables converge to perfect accuracy in the first epoch, and while lookup tables have better length-generalization, they are not as trainable as lookup tables. This experiment led to a central insight: learning tool use can be bootstrapped by swapping easily learned tools for more general tools. For instance, a model can be trained with a lookup table, which, because they have the same inputs, could be swapped for a circuit (or even a conventional calculator) for perfect generalization once parsing/selection are established.

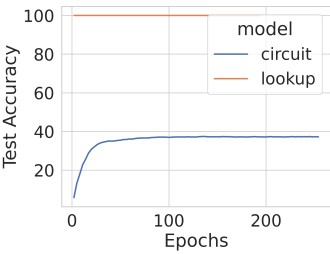

Figure 3: Circuits vs Tables

**Impact of Recursion Depth on Trainability via Fibonacci**
In this experiment, we augment a minimal neural network with a program library containing *only the recursive fibonacci function.* We use the Fibonacci numbers as a method for exploring the effect of computation depth on trainability. Specifically, we create a synthetic dataset which recursively adds numbers, given two inputs. For our study we treat this as an inherently sequential algorithm, so a recursion depth of 1 entails 8 interpreter steps, and a depth of 2 entails 16 interpreter steps and so on. Each step must be backpropagated through, potentially making gradients noisier and less stable.

| Depth | 1 | 2 | 3 | 4 | 5 | 6 |
|---|---|---|---|---|---|---|
| Interpreter Steps | 8 | 16 | 24 | 32 | 40 | 48 |
| Test Accuracy (b128) | 85.78% | 96.20% | 93.10% | 46.40% | 99.83% | 100% |
| Test Accuracy (b256) | 95.64% | 99.83% | 99.88% | 89.39% | 100% | 100% |

Table 1: Impact of Computation Depth on Trainability (Minimal Network, Base 128/256 Fibonacci)

Our hypothesis was that deeper recursions would be less trainable, however for the depths studied this effect may not have been present. Instead, it seems there were potentially statistical traps at depths 1 and 4, and these appear to have been partially mitigated by providing more training data.

In general, this preliminary study supported the idea that we could use a recurrent differentiable computer as an augmentation, despite the potential for gradient noise and numerical instability.

## 4.2 MINIMAL LLaMA WITH TOKENIZED INPUTS

Next we study the behavior of small transformers with tokenized natural language inputs. Our minimum viable transformer uses the LLaMA architecture with tiny scale parameters, and is trained from scratch. Specifically, we use the LLaMA3 tokenizer with a vocabulary of $128256$ tokens, and a scaled LLaMA with a dimension of $128$, $4$ transformer heads, $2$ key-value heads, and $4$ transformer layers. This model is trained from scratch using the AdamW optimizer [50], a batch size of $32$ and a learning rate of $1 \times 10^{-2}$.

**Parsing Modular Arithmetic from Natural Language**    We provide minimal LLaMA with a lookup table for mod-128 arithmetic operations and train it on natural language versions of one-step arithmetic, e.g. "Add 3 and 4". Solving this problem is a matter of parsing the sentence to extract the operation and operands, and then providing these to the differentiable calculator. The dataset consists of $45,151$ examples. Supervision is given only on answers, via cross-entropy. By $62$ epochs, the model can use the calculator with $99.2\%$ accuracy on the test set, and by $132$ epochs the accuracy is $100\%$. This establishes the possibility of parsing text inputs into a structure form from answer supervision alone. For example, a tokenized number is assigned a learned embedding vector, and the final layer initial register and memory values and a selected operation, in the form of one-hot encoded classifications. Accordingly, we move on to experiments with pre-trained transformers.

## 4.3 AUGMENTING LLaMA 3.2 WITH DIFFERENTIABLE MODULES

Finally, we experiment with the latest LLaMA 3.2 model, an open source transformer released by Meta. In particular, we focus on the smallest model versions with 1 billion or 3 billion parameters. Primarily, this is done in the interest of running more experiments on limited hardware, but also because we hypothesize smaller and shallower models will train more reliably than larger ones. Originally, we performed experiments with LLaMA 3.0 8B.

The augmented model is fine-tuned on a synthetic dataset, with $70\%$ of the data reserved for training. The compiled algorithms are frozen, and all the weights of the model are updated (we found fine-tuning only final layers to be less effective). We use a learning rate of $1 \times 10^{-5}$ and batch size of $2$, per Meta's recommendations for fine-tuning. In practice, we have found other hyperparameters (particularly larger batch size or learning rate) fail to converge. The primary purpose of this fine tuning is not to induce new algorithms, but to have the transformer learn which algorithms to select in which context, and what parsed inputs to provide based on a natural language sentence.

**Arithmetic**    LLMs like LLaMA have some arithmetic ability, especially with smaller numbers, but often fail on larger numbers. Important factors include tokenization and positional embeddings [51]. Like with the minimal networks before, we augmented LLaMA with a differentiable calculator and fine-tune on a large dataset. For a base-128 calculator, we find that LLaMA 3.2 1B can be fine-tuned to use a differentiable calculator perfectly, within 9 epochs. For base-256 (which has more training data), the model converges to perfect accuracy by $4$ epochs. These lookup-table based calculators are not perfectly scalable, but could be replaced with more sophisticated modules. However, even performing perfect base-256 arithmetic is sufficient for datasets like GSM-8k.

**Sorting**    Algorithms like sorting often underly other reasoning tasks, but robust sorting is difficult to learn by induction alone. We create a character-level sorting dataset, for instance "cadb" is sorted to "abcd". Language models like GPT tend to struggle on this task, as for instance they will hallucinate or forget characters, and they cannot generalize to sorting longer lists. Furthermore, tokens may contain multiple characters, making it difficult to parse such a task in the first place.

We test sorting strings of 8, 10, and 12 characters. We compile in an insertion sort algorithm to a base-128 computer. However, this algorithm is highly sequential and amplifies numerical instabilities inherent in the differentiable computer. Specifically, the algorithm is $42$ instructions long, and because of looping often requires hundreds of instructions to complete. Because the computational

model is inherently sequential, each instruction run is essentially an additional layer that must be backpropagated through.

The model is given a tokenized natural language sentence, e.g. "Sort the string "dlifdbabejld"" (Answer: "abbdddefijll"). Then, the final layer of the model outputs parsed inputs to the differentiable computer, and selects an algorithm to run. When the insertion sorting algorithm is selected, the inputs to the differentiable computer will be sorted and returned.

Despite the potential difficulties in training, a fine-tuned LLaMA 3.2 can achieve decent performance on character-level sorting when supervised only on answers.

| Size | 8 | 10 | 12 |
|---|---|---|---|
| Test Accuracy | 36.54% | 35.03% | 33.36% |

Table 2: LLaMA 3.2 1B $\Delta$ — Character Sorting

We attribute this relative lack of performance to the difficulty of parsing the problem given indirect supervision, especially with non-character-level tokenization, and plan to follow up by either pre-parsing the problem or giving problems which are unaffected by tokenization.

## 5 FINDINGS AND INSIGHTS

Despite the preliminary nature of this work, it has resulted in general insights and key challenges to solve to effectively build LLMs which are capable of reasoning.

**Parsing, Selection, Representation, and Collapse**   Fundamentally the internal algorithms and representations that an LLM or other inscrutable model learns are potentially entirely different than the ones that humans possess. The original premise of this paper was that LLM-internal representations represent the input sufficiently to the extent that they could produce parsed inputs, such as classifications of numbers or classifications of operations to perform. Then, given these representations it would be possible to select human-like algorithms from a library and provide structured inputs to them. However, it may be that the internal representations of the model are extremely different than those used by the differentiable computer, and this is likely given the theory and results in "Transformers Learn Shortcuts to Automata" [5]. Furthermore, our results highlight the overall shortcomings of gradient-descent based learning, given that even when the ideal algorithm is already present, there are still scenarios where the model may not learn a perfectly generalizable solution. This mirrors the findings in "On The Paradox of Learning to Reason from Data" [6], which compiles logical reasoning into BeRT and finds that gradient descent deviates from it immediately.

**Differentiable Computers**   The Bunel register machine is a specialized model intended to demonstrate neural compilation when it was originally published in 2016, and not necessarily optimized for scale [8]. In particular, it follows a sequential model of computation similar to its predecessors [24]. Accordingly, it may be promising to consider augmentations that use other computational models, such as a parallelized version of the Bunel model or variants of graph neural networks [32, 30, 34]. However, even augmentations like graph neural networks may have shortcomings, as they also require some sequential computation and still require inputs to be parsed into a specialized form.

**Tokenization, Embeddings, Autoregression, and Reasoning**   Large language models include design choices which seem to negatively impact reasoning ability. First, tokenization, such as that used by LLaMA 3.2, is often optimized for compressing large corpra, rather than performing certain reasoning tasks that require different tokens, such as arithmetic or letter counting. Second, positional embeddings play a large role in arithmetic and similar tasks [51]. Finally, autoregression with a shallow model does not natively allow for adaptive computation – for instance if an LLM is asked to sort a list, it may need to sort the entire list before it can output the first element [52]. Furthermore, autoregression means that a model will have to commit to mistakes if they occur early on, making generation potentially unstable.

## 6 CONCLUSION

In this study, we investigated the feasibility of augmenting large language models with libraries of differentiable programs. To some extent, differentiability is effective in assisting fine-tuning. However, there are empirical limits to the effectiveness of differentiability, especially as computational depth increases. Our experimental results establish an initial threshold for computational depths that remain trainable. Even within this limit, interesting augmentations are still possible. For example, we establish that a large language model can be fine-tuned to use a differentiable calculator effectively, and that an easily-trained calculator can be replaced by a more general one. Furthermore, we found that a large language model can be augmented with more sophisticated differentiable programs, such as an insertion sorting algorithm. However, there are barriers beyond augmentation, namely tokenization, embeddings, and autoregression, which also impact potential reasoning ability.

In future work, we plan to implement a massively parallel differentiable computer and specify a neural compiler for it. Ideally, this will be more trainable than a highly sequential model. Also, we plan to do more experiments with compositionality, which is the main motivation for neural compilation. Finally, it is unclear if the transformer architecture is truly final, given its lack of algorithmic ability. An augmented LLM may be the most feasible for short-term results, but it seems unlikely that a transformer would be sufficient for a truly general AI model.

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

## A  Appendix

## B  Numerical Representation

**One-Hot Number Encodings**   Algorithmic ability is closely tied to numeracy. However, neural networks are not natively good at representing numbers. Often, numbers are represented using unit encodings, e.g. [00100]. This can be desirable for doing dot-product based lookup, or when viewing numbers as features, but it is undesirable for scaling properties. The sparsity of these representations can be advantageous in some ways (number representations are not entangled), but disadvantageous in others (a single number provides only a single bit of supervision, and geometric distance does not correspond to number distance). Probabilistic unit encodings are calculated using a softmax layer to normalize a dense vector into a multinomial probability distribution. This can represent a wide range of other distributions.

**Binary Number Encodings**   Binary representations are highly advantageous in classical computer science, as they allow encoding an exponential amount of numbers in a linear space, e.g. for a bit vector of length $n$, we can represent numbers up to $2^n - 1$. However, binary representations may be too entangled to be used as features in neural networks, e.g. the binary encodings for 2 and 3 ([10] and [11]) overlap in the most significant bit. A unit vector has a native interpretation as a direct probability distribution over numbers, while a binary vector has a probabilistic interpretation for each bit. However, a probabilistic unit vector encodes more possible distributions than a binary one.

**Binary to Unit Conversions**   Ideally, numbers are always represented in binary, except for when they are needed as features or for differentiable indexing for lookup tables or memory. Accordingly, we wish to define bidirectional conversions between binary and unit vectors. In particular, we want to preserve the probabilistic representations of both encodings. To lookup binary vectors from unit encodings, we simply do a dot-product lookup with a $n \times b$ table of binary encodings. The reverse direction is less obvious, as we are going from $\mathbb{R}^{\log(n)}$ back to $\mathbb{R}^n$. However, it admits a closed form similar to the binomial distribution, but for independent trials. This represents, intuitively, flipping a coin at each bit to produce different numbers. For instance, for a 2-bit vector $b$, with bits $b_0$ and $b_1$, the one-hot conversion is $[(1 - b_1)(1 - b_0) \quad (1 - b_1)b_0 \quad b_1(1 - b_0) \quad b_1 b_0]$, which generalizes to higher dimensions.

## C  Technical Background

Lookup tables and memory are primarily derived from the math presented in [8] and [24, 25, 2].

**Differentiable Lookup Tables** trade memory for computation by pre-calculating the answers to input combinations. For binary operations like addition, a lookup table is a 3D tensor $M_{ijk}$ where the axes $i$ and $j$ correspond to the first operands, and $k$ is the encoding dimension of the answer. Then, a lookup is the summation $c_k = M_{ijk}a_i b_j$. Now we define a lookup table for multiple operations, e.g. the four basic arithmetic operators, or common fundamental programming operations such as max/min/inc/dec. To do so, a new leading dimension is added for the operator, so a lookup table becomes a 4D tensor $T_{hijk}$, which is indexed via three dot products, corresponding to looking up an operator, and then operands, sequentially. This is written with the Einstein summation:

$$c_k = T_{hijk}f_h a_i b_j \tag{10}$$

When memory is abundant, lookup tables are extremely favorable, as they have shallow and stable gradient paths (since they are only tensor contractions). An issue with lookup tables, and more

```
1   main:
2        set 8 r1
3        set 0 r9
4   sort:
5        copy r9 r2
6        call scan_init r8 r7
7        call swap r8 r10
8        inc r9
9        comp r1 r9 r3
10       jump r3 finish
11       set 1 r5
12       jump r5 sort # Sort the remaining array
13  swap:
14       read r9 r3   # Use r3 to store temp swapped
15       read r6 r4   # r4, value we want to swap
16       write r4 r9 # Write to r9 location
17       write r3 r6 # Write to r6 location
18       set 1 r5
19       jumpr r5 r10
20  scan_init:
21       read r2 r5
22       copy r2 r6
23  scan: # Assumes r1 holds array size
24       comp r1 r2 r3
25       jumpr r3 r7
26       read r2 r4
27       comp r4 r5 r3
28       inc r3
29       jump r3 scan_replace
30  scan_end:
31       inc r2
32       set 1 r3
33       jump r3 scan
34  scan_replace:
35       copy r4 r5
36       copy r2 r6
37       set 1 r3
38       jump r3 scan_end
39  finish:
40       halt
41       set 1 r5
42       jump r5 finish
```

Figure 4: Insertion Sort

broadly with unit vector encodings, is that they scale poorly with respect to the maximum representable number. If the maximum number is $n - 1$, then a unit vector is length $n$ (assuming zero is included). A binary-arity lookup table will be $n \times n \times n$, and a composite lookup table will be $o \times n \times n \times n$ for $o$ operations. Fundamentally, this is not scalable enough to enable arbitrary multiplication beyond very small scales, e.g. even representing a $n = 1024$ lookup table requires 32Gb of memory. This limitation is a byproduct of unit encodings. Alternatives include using binary number encodings or circuit-like representations for basic operations.

