# OpenReview forum: "Algorithmic Language Models with Neurally Compiled Libraries"
_ICLR.cc/2025/Conference — Submitted to ICLR 2025_

### Official Review · Reviewer_H5y9 · 2024-10-23

**Soundness:** 1
**Presentation:** 1
**Contribution:** 1
**Rating:** 1
**Confidence:** 4

**Summary:**

This paper trains transformers to parse textual inputs into (1) inputs of programs and (2) which program to use from the library. The program's outputs are also unparsed into texts by transformers.

Specifically, it mainly finetunes Llama (by changing its last layer) to map "Add 3 and 4" into (1) a categorical distribution over programs in the library and (2) categorical distributions over base-128 integers as the programs' inputs. The program's output is a categorical distribution over base-128 integers as well and is then unparsed into texts by neural networks.

It evaluates such a model in two tasks:
* Modular Arithmetic (providing arithmetic lookup tables in the library): 100% accuracy
* Sorting (providing insertion sort in the library): ~35% accuracy.

**Strengths:**

This paper targets an important problem, integrating programs with neural networks.

**Weaknesses:**

My main concern is about the novelty and the meaningfulness of this work:
- This paper trains/finetunes transformers to parse/unparse between texts and program inputs/outputs. I do not see the difference between this work and other LLM Tool Argumentation works. Parsing and Unparsing have become much more reliable and accurate with LLMs, let alone the problem here is just to choose one program from the library and retrieve arguments of programs from the text. Differentiable programs may provide more information for program search but they still underperform discrete-search-based methods to synthesize conventional programs. The program search problem here is again too easy to consider those nuances of different program synthesis methods...

Other concerns include
* The performance is not satisfying. For example, to learn sort with the ground-truth sort algorithm in the library, the accuracy is still lower than 40%.
* This paper is poorly written and hard to understand. After thoroughly reading the paper, I am still unsure of e.g.,
  - the components of the library, just arithmetic operators, or with sort, or also including some basic logical operators?
  - why differentiate sorting using differentiable register machines instead of many other works that potentially provide better gradient estimations?
  - Definition of test accuracies for e.g., sort;
  - Opinions/Arguments without evidence such as: "While we do not explicitly study them in this work, they almost certainly play a role."

**Questions:**

* For such a parse/unparse model, why do we need differentiable programs?
* What is the performance of baselines from the LLM Tool Argumentation field?

---

> ### Author Response · Authors · 2024-11-21
> **Response to Reviewer H5y9**
>
> We appreciate your review, but your summary suggests that you may have misread our paper. To clarify, we are using differentiable programs as a basis for training on *answers only*, not training on intermediate parsed values. In this regime, parsing may be learned implicitly, which is much more difficult than training it directly. The purpose of doing this is to test the limits of differentiability alone. To the best of our knowledge, this has not been done in other LLM Augmentation research and represents a major qualitative difference. Given this difference, we would request you to reconsider your evaluation.  We have made modifications to the writing to hopefully make the main points of our paper clearer, and hope that they help you.
>
> In the results we present, all parameters of LLaMA are trained, not just the last layer. In general we have found this to be far more effective.
>
> We agree about the strengths of discrete program search methods, however the point of this paper is not to use search for explicit programs but rather to learn how to compose existing programs for natural language tasks, something which is uniquely possible with a differentiable program library that augments an LLM. The purpose of experiments which select only a single program is to establish if the gradient signal from the differentiable computer is sufficient to do this in practice. However, experiments which compose multiple library functions are more compelling.
>
> We have removed speculative statements or supported them with experiments or citations.
>
> For calculating text accuracy, we use a binary correctness: A list is either sorted correctly or isn’t, and the accuracy is just the percentage of correctly sorted problems.
>
> > Clarify the starting library for each problem:
>
> We have updated the text to make the starting libraries clearer, but in summary we first experiment with a calculator-only augmentation, and again with an augmentation that uses a small program library. In the case of the calculator, the model classifies which operation to perform. In the case of the library, the model classifies which functions to call.
>
> > Why differentiate sorting using differentiable register machines instead of many other works that potentially provide better gradient estimations?
>
> We agree that discussing gradient estimation techniques is valuable, so we have added an appropriate section to our related work.
>
> Again, we thank you for reading and reviewing our paper, and will be happy to help in clarifying it further if needed.

---

> ### Comment · Reviewer_H5y9 · 2024-11-24
> **Clarify and suggestions**
>
> I have read the authors' responses. To clarify, I did not misunderstand the paper as the authors claimed. The current formulation of the method is still mapping input texts to programs whose execution results are mapped to texts as outputs, as I summarized. Program parsers and neural networks are two separate parts (regardless of how they are trained) and we can easily replace the parsers with either pretrained LLMs + prompts or many other traditional parsers (discrete or not) to solve the problems in this paper. That was my point.
>
> I would request the authors to check my comments again. Some quick experiments to run to make this paper more convincing, at least from my perspective:
> * Comparison with LLM Tool argument baselines, e.g., Chain-of-Code.
> * You can come up with some foreign language so that the pre-trained models won't work and joint learning would be valuable.
> * Comparison with simple parsers: Given the small library and program space, I would guess you can get pseudo-labels to train parsers by enumerating programs.
> * Evaluate the methods on more algorithmic tasks.
>
> For the future, I would suggest the authors think about how to integrate programs and neural networks more seamlessly, e.g., call programs more than once, programs calling neural networks, and non-obvious mapping from texts to programs.

---

> > ### Author Response · Authors · 2024-11-25
> > **Response to Reviewer H5y9**
> >
> > Your follow up response still does not indicate an accurate understanding of the paper (namely that the entire model is differentiable, "parsing" and program execution included). It also seems like you misunderstand the purpose of our experiments, e.g. we are not trying to solve sorting, which is obviously an already solved problem.
> >
> > We agree about including tool use baselines and appreciate the suggestion.
> >
> > The whole purpose of the paper is integrating *differentiable* programs and neural networks tightly, though there are certainly ways to expand upon this.

---

> > > ### Comment · Area_Chair_P8Xd · 2024-11-26
> > >
> > > Dear Authors, Dear Reviewer H5y9,
> > >
> > > I'm briefly chiming into this discussion as I've noticed the exchange starting to become a bit less constructive.
> > >
> > > From my read of the situation, I don't think Reviewer H5y9 misunderstood the paper. Nowhere in their two messages did they say  that the model is not end-to-end differentiable. Rather, they said:
> > >
> > > > Program parsers and neural networks are two separate parts (regardless of how they are trained)
> > >
> > > which I think is quite telling. They're making a statement about the _dataflow_ and different _components_ of a model, **not** a statement about gradient flow.
> > >
> > > The point of their suggestion seems to be that it might be useful to investigate what the performance would look like if we swapped out some of these components with pre-trained / carefully prompted LLMs that perform the same task (e.g. parsing). I think this could be a valuable experiment, because it would reveal the tradeoffs between the knowledge stored in a generalist language model vs. the specialist fine-tuning performed here.
> > >
> > > As possible inspiration for additional algorithmic evals, the Authors may consider e.g. the CLRS-Text benchmark:
> > > https://arxiv.org/abs/2406.04229 -- or any similar length generalisation oriented eval for LLMs. I'm just calling out CLRS-Text because it explicitly focusses on algorithmic routines.
> > >
> > > Best,
> > > AC

---

### Official Review · Reviewer_LpaQ · 2024-11-02

**Soundness:** 3
**Presentation:** 3
**Contribution:** 1
**Rating:** 3
**Confidence:** 5

**Summary:**

In this paper, the authors propose to augment LLMs with a differentiable computer equipped with a pre-existing library of functions, as way to to make large foundation models more capable of reliably performing classical algorithms and therefore, in the view of the authors, reasoning. This is in contrast with other tool-use or neurosymbolic approaches, in which LLMs are equipped with an external module (e.g. a calculator, or web browser, or Python interpreter) whose workings are fully interpretable, but which cannot be differentiated through.

**Strengths:**

The authors' proposal is definitely original, the paper outlines it in a mostly clear manner, and there is reason to believe that such augmentations, once refined and properly scaled, could indeed prove to be invaluable in making foundation models capable of algorithmic reasoning.

**Weaknesses:**

Ultimately, the author's proposal does not seem to work well enough given the evaluations they present, and by their own admission their paper is more of a initial proof of concept (and a limited one at that) rather than a practical demonstration of the soundness of their approach. In this regard, I cannot provide any more suggestions for improvement than the authors already do in section 6; the paper in its present state is ultimately more suited to be a workshop publication than a main conference paper.
A related point I wish to make is that the authors do not seem to address, either in the introduction or in their experiments, the issue of length generalisation, which is the main problem to solve in order to make LLMs capable of actually running algorithms rather than just find solutions via shortcuts and pattern matching. Showing that their augmented model is capable to length-generalize, even just on a very simple task such as integer sorting or parity, would significantly enhance this paper's contribution.

**Questions:**

- Am I to understand that none of the differentiable intepreter's parameters are trained? If not, which ones are?
- Related to the above, it is still not clear to me which algorithms constitute the compiled library for the experiments in section 4 and, most importantly, how are these algorithms compiled into the model before the training runs.
- In section 4.1, the authors assert that a differentiable circuit ALU generalises "beyond lookup tables". How can this be deduced by figure 3, where text accuracy for differentiable circuits is actually lower than for a lookup tables?
- How did the authors pick the training hyperparameters used in section 4.2?
- In section 4.3, to the authors consider generalising to sorting longer lists at test time (e.g., training on list of size 8 and testing on strings of size 12)
- Again in section 4.3, the authors attribute the weak performance of their augmented model to the difficulty of parsing the natural language description of the problem. Can they provide any evidence (e.g. in the appendix) for this statement via some ablation?

---

> ### Author Response · Authors · 2024-11-21
> **Response to Reviewer LpaQ**
>
> > The authors' proposal is definitely original, the paper outlines it in a mostly clear manner, and there is reason to believe that such augmentations, once refined and properly scaled, could indeed prove to be invaluable in making foundation models capable of algorithmic reasoning.
>
> We greatly appreciate the reviewer’s assessment, and while we acknowledge the preliminary nature of the work, we also believe that, even in the current state, there is significant value to the research community in the sense that other research work may be able to build on what we have done so far.
>
> We'd like to point out that the main point of our paper is in identifying precisely how such techniques could be refined and scaled -- in particular, we establish/confirm that gradient descent based training is highly reliant on shallow and highly parallel gradient paths that are less noisy, which suggests that our approach would work far better if done with a highly parallel differentiable computer rather than the sequential one we have opted for. Accordingly, we believe the paper has value to the ICLR community in the current form, in that it gives clear next steps for this sub-field.
>
> > In this regard, I cannot provide any more suggestions for improvement than the authors already do in section 6
>
> Indeed, we are well-aware of the potential ways this paper could be improved, and hope to make significant progress in doing so. At the same time, some limitations are fundamental (e.g. sequential computation <> differentiability), while other limitations are out of scope of a single paper (e.g. numeric representations, tokenization). With this in mind, we find it more appropriate to publish our intermediate work in the interest of exploring other aspects of this problem, namely highly parallel differentiable computing.
>
> > Length generalization
>
> We agree with the reviewers assessment and highlight length generalization is an essential reason why neurally compiled libraries would be desirable. We may update the paper with results in length-generalization
>
> > Am I to understand that none of the differentiable intepreter's parameters are trained? If not, which ones are?
>
> Yes, it is accurate to say that the differentiable interpreter is not trained, as in general it is not parameterized. Technically the program library itself and aspects such as lookup tables *could* be parameterized, but we opt for them to remain static since this maintains guarantees of their functionality. Accordingly, the only components that are trained are the transformer itself and specialized classification layers that convert the final embedding into classifications of inputs and selected programs.
>
> > Related to the above, it is still not clear to me which algorithms constitute the compiled library for the experiments in section 4 and, most importantly, how are these algorithms compiled into the model before the training runs.
>
> We have updated the manuscript to clarify what the initial library looks like in each experiment.
>
> > In section 4.1, the authors assert that a differentiable circuit ALU generalises "beyond lookup tables". How can this be deduced by figure 3, where text accuracy for differentiable circuits is actually lower than for a lookup tables?
>
> The text accuracy is a side-effect of trained parsing ability, not the augmentation. When we say that a circuit generalizes further, we are actually talking about length generalization! A table generalizes only to a fixed length (it only supports modular arithmetic), but a circuit can generalize to arbitrarily long numbers without modification (if/when the inputs are parsed correctly and the operator is chosen correctly!).
>
> > How did the authors pick the training hyperparameters used in section 4.2?
>
> The hyperparameters used are based on Meta’s recommendations for fine-tuning, as well as manual experimentation that confirmed these as effective choices.
>
> > In section 4.3, to the authors consider generalising to sorting longer lists at test time (e.g., training on list of size 8 and testing on strings of size 12)
>
> This is a valuable suggestion.
>
> > Again in section 4.3, the authors attribute the weak performance of their augmented model to the difficulty of parsing the natural language description of the problem. Can they provide any evidence (e.g. in the appendix) for this statement via some ablation?
>
> We are currently running experiments which do sorting on prompts that are less sensitive to external factors, e.g. tokenization.
>
> We thank you again for your diligent review and questions.

---

> > ### Comment · Reviewer_LpaQ · 2024-11-26
> >
> > I have read the authors' response, as well as parsed the changed they made in their paper revision.
> > I thanks the authors for their response to my questions and their revision of the paper. However, these interventions do not ultimately change the fact that this work is preliminary and exploratory in nature, an assessment that the authors themselves agree with.
> > Given this, while I do agree that the authors' observation and insights are useful to the community, and that the authors are likely on the right path to a major contribution ("a massively parallel differentiable computer and specify a neural compiler for it"), I  believe that, until they get there, the contributions in the present work are not strong enough for publication in ICLR. I therefore leave my score unchanged.

---

> > > ### Author Response · Authors · 2024-11-26
> > > **Thanks to Reviewer LpaQ**
> > >
> > > Thank you for your response and thoughtful review!

---

### Official Review · Reviewer_VhLJ · 2024-11-03

**Soundness:** 2
**Presentation:** 2
**Contribution:** 3
**Rating:** 6
**Confidence:** 3

**Summary:**

The authors investigated the feasibility of augmenting large language models with libraries of
differentiable programs.  This is important and very interesting direction.
They augment LLaMA 3.2 with a differentiable interpreter,  develop differentiable algorithm library,
and study how a model can utilise given functions.  This could be potentially great work, but lack of experimentation makes it less appealing.

**Strengths:**

- Reasoning, arithmetic and algorithmic abilities are still weak spot of LLM .  'Algorithmic Language Models with Neurally Compiled Libraries' suggest interesting and promising approach to improve capabilities of LLM.
- Authors analyse impact of recursion depth on trainability on Fibonacci dataset
- They create library of differentiable modules and augment LLM with them

**Weaknesses:**

- Neurally Compiled Library is primary experimental work. Therefore I believe work would greatly benefit from extending it's evaluation on more, preferable public datasets. Also more detailed about experiments performed (for example what dataset was used for Airithmetic testing on page 8)  would make it more interesting.
- It would be helpful to have baselines with/without  differential modules (i.e Figure 3,  Table 2)
- Model background section would benefit from either added citations or clear indication what was proposed in current work. (I.e Differentiable Registers, Probabilistic execution, Differentiable Interpreter sections)

**Questions:**

On page 4  'instruction counter' and  'program counter' are mentioned. Could you please clarify what is the difference?
Do formulae (3) and (4) assume broadcasting?
On page 8.  There is statement 'fine-tuned LLaMA 3.2 can achieve descent performance'. Could you please clarify, what you comparing agains?
'Furthermore, our results highlight the overall shortcomings of gradient-descent based learning, given that even when the ideal algorithm is already present, there are still scenarios where the model may not learn a perfectly generalizable solution.' This is very interesting finding.  I think both paper and community would benefit if more details on training difficulties, what didn't work, etc would be given.

---

> ### Author Response · Authors · 2024-11-21
> **Response to Reviewer VhLJ**
>
> Thank you for recognizing our direction as important and interesting. We understand why you may conclude our work is preliminary, but hope to convince you it has value to the ICLR community in the current form.
>
> > Reasoning, arithmetic and algorithmic abilities are still weak spot of LLM . 'Algorithmic Language Models with Neurally Compiled Libraries' suggest interesting and promising approach to improve capabilities of LLM.
>
> This is precisely the point of the paper -- reasoning abilities in LLMs are currently lacking, and will rely on more fundamental changes, namely algorithmic ability, to improve. LLM reasoning is one of the most important research topics currently, and we hope our paper progresses this line of research in a unique way.
>
> > Neurally Compiled Library is primary experimental work. Therefore I believe work would greatly benefit from extending it's evaluation on more, preferable public datasets.
>
> We agree about evaluating on public datasets. In particular, we would like to extend our evaluation to the CLRS benchmark [1]. In the current form, we use synthetic datasets which are essentially a subset of CLRS.
>
> We plan to add baselines before the end of the review period.
>
> > Model background section would benefit from either added citations or clear indication what was proposed in current work. (I.e Differentiable Registers, Probabilistic execution, Differentiable Interpreter sections)
>
> We have further modified the methods section of the paper to make it clearer what is being introduced in this work. Also while it is fair to say that the paper is heavily reliant on methods introduced in Bunel 2016, we’d like to point out the critical difference that Bunel 2016 focused on learning contextual programs (e.g. that perform better on biased data), while we focus on using unaltered programs as a method for augmenting LLMs. So while the math is very similar, there is a significant qualitative difference in the purpose it is used for (copied from response to reviewer EFCQ).
>
> > On page 4 'instruction counter' and 'program counter' are mentioned. Could you please clarify what is the difference?
>
> We have used these terms interchangably, but have updated the paper to prefer 'instruction counter', which is more accurate.
>
> > Discussion of sorting performance
>
> Please see our overall response, however we believe the problem of learning to parse/select the differentiable insertion sorting algorithm from answer supervision alone is quite difficult. Regardless, we plan to provide baselines for comparison and ideally an ablation study testing if our intuitions about why performance is bad (e.g. tokenization) are correct.
>
> Furthermore, we do not really expect the differentiable insertion sort algorithm to succeed, because the gradient path produced by the algorithm is so long and noisy. We use this as evidence for our main insight, namely that a differentiable augmentation of an LLM should focus on highly parallel algorithms.
>
> > our results highlight the overall shortcomings of gradient-descent based learning, given that even when the ideal algorithm is already present, there are still scenarios where the model may not learn a perfectly generalizable solution.' This is very interesting finding. I think both paper and community would benefit if more details on training difficulties, what didn't work, etc would be given.
>
> We agree, and will expand on this in our paper. Please see [2] which has a very similar finding.
>
> [1] Veličković, P., Badia, A. P., Budden, D., Pascanu, R., Banino, A., Dashevskiy, M., ... & Blundell, C. (2022, June). The CLRS algorithmic reasoning benchmark. In International Conference on Machine Learning (pp. 22084-22102). PMLR.
>
> [2] Zhang, H., Li, L. H., Meng, T., Chang, K. W., & Broeck, G. V. D. (2022). On the paradox of learning to reason from data. arXiv preprint arXiv:2205.11502.

---

### Official Review · Reviewer_EFCQ · 2024-11-04

**Soundness:** 1
**Presentation:** 2
**Contribution:** 2
**Rating:** 5
**Confidence:** 3

**Summary:**

The paper addresses LLMs’ problem with performing symbolic operations. To this end, they investigate one way to incorporate a differentiable interpreter into LLMs. Given a text input, the authors use LLaMa 3.2 to select the correct program, out of a library of programs, and to generate the program’s inputs. A differentiable interpreter then runs the program. A final neural network is then used to produce the final output.
The paper presents experiments on arithmetic and sorting, as well as ablation experiments with simpler neural network models.

**Strengths:**

Address an important problem, which is well motivated.
The background section is comprehensive and an interesting read.
It proposes an interesting approach of using a differentiable interpreter, as well as preparing a library of programs to choose from, by compiling symbolic programs into differentiable versions.

**Weaknesses:**

Poor introduction: most of the introduction, apart from the very last paragraph is dedicated to motivating the work. The very last paragraph has 1 sentence which describes the methodology.

Section 3, Methodology: The authors should make it clear what their contributions are. I am left with the impression that the majority of this section (apart from 3.4) are ideas from a previous paper that are just re-stated here. If this is the case, it should be stated more clearly. In itself, 3.4 is very brief and doesn’t describe the method sufficiently well, for example, I am unsure if the method selects only a single program or runs multiple programs during training.

Experiment section, poor presentation:  the setup of each experiment isn’t described clearly: I am unsure what is the set of programs considered, what the neural architecture is for 4.1 and 4.2, what the inputs and outputs look like, what’s the difference between circuits and tables

Experiment section, no baselines: there are no purely neural baselines. Section 4.2 augments and finetunes LLaMa without showing LLaMa’s performance.

Experiment section, unconvincing results: It is not clear to me that the results support the claim in the Introduction that “resulting in a model which is universally expressive, adaptive, and interpretable”.  Specifically, Table 2 presents the result for sorting, where the accuracy is between 33% and 37% which the authors refer to as “decent performance”. I cannot see a way to reaching this conclusion. This leaves an impression that, while the method could perform well in the future, is currently underperforming and unconvincing.

**Questions:**

How do you differentiate between the selection of different programs? Do you run all of them at once? If not, how do you expect the LLM to learn which program to select if it’s trained only on the error of the outputs?

What are the novel ideas which one can take away from the methodology of this paper?

How do the experiments demonstrate that your method is “universally expressive, adaptive, and interpretable”?

---

> ### Author Response · Authors · 2024-11-21
> **Response to Reviewer EFCQ**
>
> Thank you for your diligent and thoughtful review of our paper.
>
> > Poor introduction: most of the introduction, apart from the very last paragraph, is dedicated to motivating the work. The very last paragraph has 1 sentence which describes the methodology.
>
> We have modified the last paragraph of our introduction to be dedicated to describing the methodology at a high level. At the same time, we believe the motivation sections are very important, as it seems to have sold all four reviewers on the premise of our paper.
>
> > Clarify contributions
>
> We have edited several sections of the methodology section to make it clearer what our contributions are. Also, we have significantly expanded section 3.4. While it is fair to say that the paper is heavily reliant on methods introduced in Bunel 2016, we’d like to point out the critical difference that Bunel 2016 focused on learning contextual programs (e.g. that perform better on biased data), while we focus on using unaltered programs as a method for augmenting LLMs. So while the math is very similar, there is a significant qualitative difference in the purpose it is used for.
>
> > Clarify experimental setups
>
> In 4.1, 4.2, and 4.3, we have added more detail regarding which libraries were used as input. We believe there is sufficient detail regarding the neural architectures (e.g. "Our minimum viable transformer uses the LLaMA architecture with tiny scale parameters.. a scaled LLaMA with a dimension of 128, 4 transformer heads, 2 key-value heads, and 4 transformer layers. This model is trained from scratch using the AdamW optimizer [50], a batch size of 32 and a learning rate of 1 × 10−2"). However, we will add more detail to the appendix.
>
> > baselines and results
>
> We plan to add more baselines before the end of the review period, and have clarified our claims relative to our empirical results. Please also see the overall response.
>
> > How do you differentiate between the selection of different programs? Do you run all of them at once? If not, how do you expect the LLM to learn which program to select if it’s trained only on the error of the outputs?
>
> In the cases where there are multiple functions available in the library, the model selects one of these functions as a softmax classification, and effectively all available functions are run at once. In principle this allows the model to potentially learn selection from only answer supervision (which it is able to do for arithmetic for instance, where it learns to classify operators without operator supervision), however it is also feasible that some intermediates, e.g. selected library functions, could be supervised.
>
> > What are the novel ideas which one can take away from the methodology of this paper?
>
> Augmenting a LLM with a neurally compiled library has never been tried before, and to our knowledge this is the first attempt at doing so, and compared to similar approaches (e.g. [1]), represents an advance in the state of the art.
>
> Beyond the novelty of the methods themselves, there are several insights to be gained from the work, even in its preliminary form. We discuss these in the final section of our paper, but one that stands out is: differentiable augmentations will be more effective for highly-parallel and shallow programs, e.g. future work should focus on machines/algorithms that exploit this, rather than a sequential RNN-like augmentation that we attempt.
>
> [1] Bounsi, W., Ibarz, B., Dudzik, A., Hamrick, J. B., Markeeva, L., Vitvitskyi, A., ... & Veličković, P. (2024). Transformers meet Neural Algorithmic Reasoners. arXiv preprint arXiv:2406.09308.

---

> > ### Comment · Reviewer_EFCQ · 2024-11-28
> >
> > Thank your for your reply. I think the new version of the paper is an improvement. As a result, I've increased my score. Still, I am hesitant to increase the score further, given the limited amount of baselines and the results on sorting. At present, while the idea is interesting, the paper fails to convince me of its promise.

---

### Author Response · Authors · 2024-11-21
**Overall Response to Reviewers**

We greatly appreciate the diligent effort and constructive nature of all four reviewers, and thank them for their time. All four reviewers agree that the work is overall “well motivated”, “an important and interesting direction,” and “original, clear and .. could prove invaluable in making foundation models capable of algorithmic reasoning”. However, all four reviewers also mention that the work is preliminary, especially when it comes to positive experimental results. While we see the reasoning for this assessment, we have expanded the experiments, and believe the work has significant value to the research community as it stands. In particular, we’d like to highlight that certain experiments, such as recursive fibonacci or differentiable insertion sort are intended to demonstrate the limits of differentiable computing, namely that differentiability is most effective for shallow and highly parallel algorithms, but struggles as computational depth increases. While this is expected, this insight/confirmation will shape our future work, namely into highly parallel differentiable computers.


We would like to also highlight that, to our knowledge, the idea of augmenting an LLM with a neurally compiled library is entirely novel and has benefits over the current state-of-the-art in algorithmic reasoning. We discuss the most closely-related paper (Transformers Meet Neural Algorithmic Reasoners [1]). In comparison to this work, we are actually attempting to solve a more difficult problem, and do so more efficiently. In particular, the symbolic inputs are not pre-parsed into graph form, and in our setup, parsing must be learned implicitly, which is exceptionally difficult! Beyond this, instead of approximating an algorithm via supervised learning (which requires significant computation and does not carry guarantees), we use neural compilation to create a perfect differentiable version of the algorithm, which provides guarantees and is efficient because it does not involve training to create the algorithmic module.  Because of the ambitious nature of this problem setup (learning parsing/selection from final inputs alone), we consider the sorting results promising and demonstrative even if they are very underwhelming for any practical application.

Beyond a scenario where a single algorithm is selected and used, our ambition was to provide a diverse library of programs that could be selected and composed to be used in more difficult problems. We believe that this potential for composing algorithms is the ideal way to solve the issue of algorithmic reasoning ability in large language models, which is arguably one of the most important contemporary problems in our field.

Again, we greatly appreciate the effort given by all reviewers, and hope you are convinced that our work, while preliminary, offers value to the ICLR research community.

[1] Bounsi, W., Ibarz, B., Dudzik, A., Hamrick, J. B., Markeeva, L., Vitvitskyi, A., ... & Veličković, P. (2024). Transformers meet Neural Algorithmic Reasoners. arXiv preprint arXiv:2406.09308.

---

### Meta-Review · Area_Chair_P8Xd · 2024-12-21

**Metareview:**

This is a very interesting paper leveraging pre-compiled differentiable neural libraries as part of a language model architecture. In my opinion, this is definitely an idea worth pursuing further. However, it is also evident that the paper's evaluation is still preliminary, with the majority of the Reviewers still voting to reject the paper. I will concur with them and recommend rejection now, but strongly encourage the Authors to pursue this research further and resubmit at the next suitable opportunity!

**Additional Comments On Reviewer Discussion:**

I believe that the discussion between Authors and Reviewers has been on the whole constructive, insofar as there were many useful suggestions raised for how to improve the paper's evaluation, and the Authors acknowledged a lot of these as valuable: for example, more elaborate evaluation under input distribution shifts, or more detailed ablations on the utility of tuned LLMs in the proposed pipeline.

Should the Authors move to address most of these, in my opinion, it should be a paper ready for publication in the next cycle.

Lastly, just for completeness: I have not taken into account Reviewer H5y9's score reduction to 1 when making this meta-review. I do not think this reduction was relevant to their paper _or_ the rebuttal's assessment.

---

### Decision · Program_Chairs · 2025-01-22

Reject